# LARGE ASSOCIATIVE MEMORY PROBLEM IN NEUROBIOLOGY AND MACHINE LEARNING

**Dmitry Krotov**
MIT-IBM Watson AI Lab
IBM Research
krotov@ibm.com

**John Hopfield**
Princeton Neuroscience Institute
Princeton University
hopfield@princeton.edu

## ABSTRACT

Dense Associative Memories or modern Hopfield networks permit storage and reliable retrieval of an exponentially large (in the dimension of feature space) number of memories. At the same time, their naive implementation is non-biological, since it seemingly requires the existence of many-body synaptic junctions between the neurons. We show that these models are effective descriptions of a more microscopic (written in terms of biological degrees of freedom) theory that has additional (hidden) neurons and only requires two-body interactions between them. For this reason our proposed microscopic theory is a valid model of large associative memory with a degree of biological plausibility. The dynamics of our network and its reduced dimensional equivalent both minimize energy (Lyapunov) functions. When certain dynamical variables (hidden neurons) are integrated out from our microscopic theory, one can recover many of the models that were previously discussed in the literature, e.g. the model presented in "Hopfield Networks is All You Need" paper. We also provide an alternative derivation of the energy function and the update rule proposed in the aforementioned paper and clarify the relationships between various models of this class.

## 1 INTRODUCTION

Associative memory is defined in psychology as the ability to remember (link) many sets, called memories, of unrelated items. Prompted by a large enough subset of items taken from one memory, an animal or computer with an associative memory can retrieve the rest of the items belonging to that memory. The diverse human cognitive abilities which involve making appropriate responses to stimulus patterns can often be understood as the operation of an associative memory, with the "memories" often being distillations and consolidations of multiple experiences rather than merely corresponding to a single event.

The intuitive idea of associative memory can be described using a "feature space". In a mathematical model abstracted from neurobiology, the presence (or absence) of each particular feature $i$ is denoted by the activity (or lack of activity) of a model neuron $v_i$ due to being directly driven by a feature signal. If there are $N_f$ possible features, there can be only at most $N_f^2$ distinct connections (synapses) in a neural circuit involving only these neurons. Typical cortical synapses are not highly reliable, and can store only a few bits of information[1]. The description of a particular memory requires roughly $N_f$ bits of information. Such a system can therefore store at most $\sim N_f$ unrelated memories. Artificial neural network models of associative memory (based on attractor dynamics of feature neurons and understood through an energy function) exhibit this limitation even with precise synapses, with limits of memory storage to less than $\sim 0.14 N_f$ memories (Hopfield, 1982).

---

[1]For instance, a recent study (Bromer et al., 2018) reports the information content of individual synapses ranging between $2.7$ and $4.7$ bits, based on electron microscopy imaging, see also (Bartol Jr et al., 2015). These numbers refer to the structural accuracy of synapses. There is also electrical and chemical noise in synaptic currents induced by the biophysical details of vesicle release and neurotransmitter binding. The unreliability of the fusion of pre-synaptic vesicles (containing neurotransmitter) with the pre-synaptic neuron membrane is the dominant source of trial-to-trial synaptic current variation (Allen & Stevens, 1994). This noise decreases the electrical information capacity of individual synapses from the maximal value that the synaptic structure would otherwise provide.

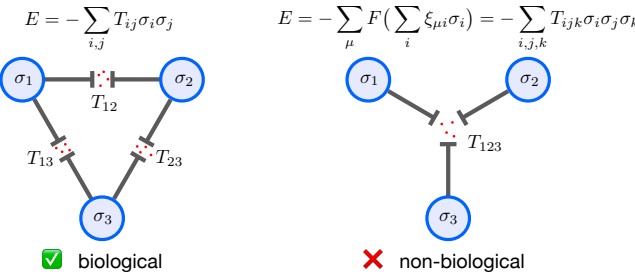

Figure 1: Two binary networks consisting of three neurons $\sigma_1, \sigma_2, \sigma_3 = \{\pm 1\}$. On the left is the classical Hopfield network (Hopfield, 1982) with the matrix $T_{ij} = \sum_\mu \xi_{\mu i} \xi_{\mu j}$ being the outer product of memory vectors (see section 2 for the definitions of notations). In this case the matrix $T_{ij}$ is interpreted as a matrix of synaptic connections between cells $i$ and $j$. On the right is a Dense Associative Memory network of (Krotov & Hopfield, 2016) with cubic interaction term $F(x) = x^3$. In this case the corresponding tensor $T_{ijk} = \sum_\mu \xi_{\mu i} \xi_{\mu j} \xi_{\mu k}$ has three indices, thus cannot be interpreted as a biological synapse, which can only connect two cells.

Situations arise in which the number $N_f$ is small and the desired number of memories far exceeds $\sim N_f$, see some examples from biological and AI systems in Section 4. In these situations the associative memory model of (Hopfield, 1982) would be insufficient, since it would not be able to memorize the required number of patterns. At the same time, models of associative memory with large storage capacity considered in our paper, can easily solve these problems.

The starting point of this paper is a machine learning approach to associative memory based on an energy function and attractor dynamics in the space of $N_f$ variables, called Dense Associative Memory (Krotov & Hopfield, 2016). This idea has been shown to dramatically increase the memory storage capacity of the corresponding neural network (Krotov & Hopfield, 2016; Demircigil et al., 2017) and was proposed to be useful for increasing robustness of neural networks to adversarial attacks (Krotov & Hopfield, 2018). Recently, an extension of this idea to continuous variables, called modern Hopfield network, demonstrated remarkably successful results on the immune repertoire classification (Widrich et al., 2020), and provided valuable insights into the properties of attention heads in Transformer architectures (Ramsauer et al., 2020).

Dense Associative Memories or modern Hopfield networks, however, cannot describe biological neural networks in terms of true microscopic degrees of freedom, since they contain many-body interaction terms in equations describing their dynamics and the corresponding energy functions. To illustrate this point consider two networks: a conventional Hopfield network (Hopfield, 1982) and a Dense Associative Memory with cubic interaction term in the energy function (see Fig. 1). In the conventional network the dynamics is encoded in the matrix $T_{ij}$, which represents the strengths of the synaptic connections between feature neurons $i$ and $j$. Thus, this network is manifestly describable in terms of only two-body synapses, which is approximately true for many biological synapses. In contrast, a Dense Associative Memory network with cubic energy function naively requires the synaptic connections to be tensors $T_{ijk}$ with three indices, which are harder, although not impossible, to implement biologically. Many-body synapses become even more problematic in situations when the interaction term is described by a more complicated function than a simple power (in this case the Taylor expansion of that function would generate a series of terms with increasing powers).

Many-body synapses typically appear in situations when one starts with a microscopic theory described by only two-body synapses and integrates out some of the degrees of freedom (hidden neurons). The argument described above based on counting the information stored in synapses in conjunction with the fact that modern Hopfield nets and Dense Associative Memories can have a huge storage capacity hints at the same solution. The reason why these networks have a storage capacity much greater than $N_f$ is because they do not describe the dynamics of only $N_f$ neurons, but rather involve additional neurons and synapses.

Thus, there remains a theoretical question: what does this hidden circuitry look like? Is it possible to introduce a set of hidden neurons with appropriately chosen interaction terms and activation

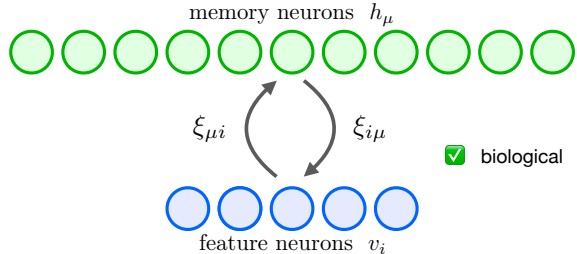

Figure 2: An example of a continuous network with $N_f = 5$ feature neurons and $N_h = 11$ complex memory (hidden) neurons with symmetric synaptic connections between them.

functions so that the resulting theory has both large memory storage capacity (significantly bigger than $N_f$), and, at the same time, is manifestly describable in terms on only two-body synapses?

The main contributions of this current paper are the following. First, we extend the model of (Krotov & Hopfield, 2016) to continuous state variables and continuous time, so that the state of the network is described by a system of non-linear differential equations. Second, we couple an additional set of $N_h$ "complex neurons" or "memory neurons" or hidden neurons to the $N_f$ feature neurons. When the synaptic couplings and neuron activation functions are appropriately chosen, this dynamical system in $N_f + N_h$ variables has an energy function describing its dynamics. The minima (stable points) of this dynamics are at the same locations in the $N_f$ - dimensional feature subspace as the minima in the corresponding Dense Associative Memory system. Importantly, the resulting dynamical system has a mathematical structure of a conventional recurrent neural network, in which the neurons interact only in pairs through a two-body matrix of synaptic connections. We study three limiting cases of this new theory, which we call models A, B, and C. In one limit (model A) it reduces to Dense Associative Memory model of (Krotov & Hopfield, 2016) or (Demircigil et al., 2017) depending on the choice of the activation function. In another limit (model B) our model reduces to the network of (Ramsauer et al., 2020). Finally, we present a third limit (model C) which we call Spherical Memory model. To the best of our knowledge this model has not been studied in the literature. However, it has a high degree of symmetry and for this reason might be useful for future explorations of various models of large associative memory and recurrent neural networks in machine learning.

For the purposes of this paper we defined "biological plausiblity" as the absence of many-body synapses. It is important to note that there other aspects in which our model described by equations (1) below is biologically implausible. For instance, it assumes that the strengths of two physically different synapses $\mu \rightarrow i$ and $i \rightarrow \mu$ are equal. This assumption is necessary for the existence of the energy function, which makes it easy to prove the convergence to a fixed point. It can be relaxed in equations (1), which makes them even more biological, but, at the same time, more difficult to analyse.

## 2 Mathematical Formulation

In this section, we present a simple mathematical model in continuous time, which, on one hand, permits the storage of a huge number of patterns in the artificial neural network, and, at the same time, involves only pairwise interactions between the neurons through synaptic junctions. Thus, this system has the useful associative memory properties of the AI system, while maintaining conventional neural network dynamics and thus a degree of biological plausibility.

The spikes of action potentials in a pre-synaptic cell produce input currents into a postsynaptic neuron. As a result of a single spike in the pre-synaptic cell the current in the post-synaptic neuron rises instantaneously and then falls off exponentially with a time constant $\tau$. In the following the currents of the feature neurons are denoted by $v_i$ (which are enumerated by the latin indices), and the currents of the complex memory neurons are denoted by $h_\mu$ ($h$ stands for hidden neurons, which are enumerated by the greek indices). A simple cartoon of the network that we discuss is shown in Fig.2. There are no synaptic connections among the feature neurons or the memory neurons. A matrix $\xi_{\mu i}$ denotes the strength of synapses from a feature neuron $i$ to the memory neuron $\mu$. The synapses are assumed to be symmetric, so that the same value $\xi_{i\mu} = \xi_{\mu i}$ characterizes a different physical

synapse from the memory neuron $\mu$ to the feature neuron $i$. The outputs of the memory neurons and the feature neurons are denoted by $f_\mu$ and $g_i$, which are non-linear functions of the corresponding currents. In some situations (model A) these outputs can be interpreted as activation functions for the corresponding neurons, so that $f_\mu = f(h_\mu)$ and $g_i = g(v_i)$ with some non-linear functions $f(x)$ and $g(x)$. In other cases (models B and C) these outputs involve contrastive normalization, e.g. a softmax, and can depend on the currents of all the neurons in that layer. In these cases $f_\mu = f(\{h_\mu\})$ and $g_i = g(\{v_i\})$. For the most part of this paper one can think about them as firing rates of the corresponding neurons. In some limiting cases, however, the function $g(v_i)$ will have both positive and negative signs. Then it should be interpreted as the input current from a pre-synaptic neuron. The functions $f(h_\mu)$ and $g(v_i)$ are the only nonlinearities that appear in our model. Finally, the time constants for the two groups of neurons are denoted by $\tau_f$ and $\tau_h$. With these notations our model can be written as

$$
\begin{cases}
\tau_f \frac{dv_i}{dt} = \sum_{\mu=1}^{N_h} \xi_{i\mu} f_\mu - v_i + I_i \\
\tau_h \frac{dh_\mu}{dt} = \sum_{i=1}^{N_f} \xi_{\mu i} g_i - h_\mu
\end{cases}
\tag{1}
$$

where $I_i$ denotes the input current into the feature neurons.

The connectivity of our network has the structure of a bipartite graph, so that the connections exist between two groups of neurons, but not within each of the two groups. This design of a neural network is inspired by the class of models called Restricted Boltzmann Machines (RBM) (Smolensky, 1986). There is a body of literature studying thermodynamic properties of these systems and learning rules for the synaptic weights. In contrast, the goal of our work is to write down a general dynamical system and an energy function so that the network has useful properties of associative memories with a large memory storage capacity, is describable only in terms of manifestly two-body synapses, and is sufficiently general so that it can be reduced to various models of this class previously discussed in the literature. We also note that although we use the notation $v_i$ ($v$ stands for visible neurons), commonly used in the RBM literature, it is more appropriate to think about $v_i$ as higher level features. For example the input to our network can be a latent representation produced by a convolutional neural network or a latent representation of a BERT-like system (Devlin et al., 2018) rather than raw input data. Additionally, our general formulation makes it possible to use a much broader class of activation functions (e.g. involving contrastive or spherical normalization) than those typically used in the RBM literature. Also, the relationship between Dense Associative Memories and RBMs has been previously studied in (Barra et al., 2018; Agliari & De Marzo, 2020). We also note that a Hopfield network with exponential capacity was studied in (Chaudhuri & Fiete, 2019), but their construction requires specifically engineered memory vectors and cannot be applied to general arbitrary memory vectors.

Mathematically, equations (1) describe temporal evolution of two groups of neurons. For each neuron its temporal updates are determined by the inputs from other neurons and its own state (the decay term on the right hand side of the dynamical equations). For this reason, an energy function for this system is expected to be represented as a sum of three terms: two terms describing the neurons in each specific group, and the interaction term between the two groups of neurons. We have chosen the specific mathematical form of these three terms so that the energy function decreases on the dynamical trajectory. With these choices the energy function for the network (1) can be written as

$$
E(t) = \Big[ \sum_{i=1}^{N_f} (v_i - I_i) g_i - L_v \Big] + \Big[ \sum_{\mu=1}^{N_h} h_\mu f_\mu - L_h \Big] - \sum_{\mu, i} f_\mu \xi_{\mu i} g_i
\tag{2}
$$

Here we introduced two Lagrangian functions $L_v(\{v_i\})$ and $L_h(\{h_\mu\})$ for the feature and the hidden neurons. They are defined through the following equations, so that derivatives of the Lagrangian functions correspond to the outputs of neurons

$$
f_\mu = \frac{\partial L_h}{\partial h_\mu}, \quad \text{and} \quad g_i = \frac{\partial L_v}{\partial v_i}
\tag{3}
$$

With these notations expressions in the square brackets in (2) have a familiar from classical mechanics structure of the Legendre transform between a Lagrangian and an energy function. By taking time derivative of the energy and using dynamical equations (1) one can show (see Appendix A for

details) that the energy monotonically decreases on the dynamical trajectory

$$\frac{dE(t)}{dt} = -\tau_f \sum_{i,j=1}^{N_f} \frac{dv_i}{dt} \frac{\partial^2 L_v}{\partial v_i \partial v_j} \frac{dv_j}{dt} - \tau_h \sum_{\mu,\nu=1}^{N_h} \frac{dh_\mu}{dt} \frac{\partial^2 L_h}{\partial h_\mu \partial h_\nu} \frac{dh_\nu}{dt} \leq 0 \tag{4}$$

The last inequality sign holds provided that the Hessian matrices of the Lagrangian functions are positive semi-definite.

In addition to decrease of the energy function on the dynamical trajectory it is important to check that for a specific choice of the activation functions (or Lagrangian functions) the corresponding energy is bounded from below. This can be achieved for example by using bounded activation function for the feature neurons $g(v_i)$, e.g. hyperbolic tangent or a sigmoid. Provided that the energy is bounded, the dynamics of the neural network will eventually reach a fixed point, which corresponds to one of the local minima of the energy function[2].

The proposed energy function has three terms in it: the first term depends only on the feature neurons, the second term depends only on the hidden neurons, and the third term is the "interaction" term between the two groups of neurons. Note, that this third term is manifestly describable by two-body synapses - a function of the activity of the feature neurons is coupled to another function of the activity of the memory neurons, and the strength of this coupling is characterized by the parameters $\xi_{\mu i}$. The absence of many-body interaction terms in the energy function results in the conventional structure (with unconventional activation functions) of the dynamical equations (1). Each neuron collects outputs of other neurons, weights them with coefficients $\xi$ and generates its own output. Thus, the network described by equations (1) is biologically plausible according to our definition (see Introduction).

Lastly, note that the memory patterns $\xi_{\mu i}$ of our network (1) can be interpreted as the strengths of the synapses connecting feature and memory neurons. This interpretation is different from the conventional interpretation, in which the strengths of the synapses is determined by matrices $T_{ij} = \sum_\mu \xi_{\mu i} \xi_{\mu j}$ (see Fig. 1), which are outer products of the memory vectors (or higher order generalizations of the outer products).

## 3 EFFECTIVE THEORY FOR FEATURE NEURONS

In this section we start with the general theory proposed in the previous section and integrate out hidden neurons. We show that depending on the choice of the activation functions this general theory reduces to some of the models of associative memory previously studied in the literature, such as classical Hopfield networks, Dense Associative Memories, and modern Hopfield networks. The update rule in the latter case has the same mathematical structure as the dot-product attention (Bahdanau et al., 2014) and is also used in Transformer networks (Vaswani et al., 2017).

### 3.1 MODEL A. DENSE ASSOCIATIVE MEMORY LIMIT.

Consider the situation when the dynamics of memory neurons $h_\mu$ is fast. Mathematically this corresponds to the limit $\tau_h \to 0$. In this case the second equation in (1) equilibrates quickly, and can be solved as

$$h_\mu = \sum_{i=1}^{N_f} \xi_{\mu i} g_i \tag{5}$$

Additionally, assume that the Lagrangian functions for the feature and the memory neurons are additive for individual neurons

$$L_h = \sum_\mu F(h_\mu), \quad \text{and} \quad L_v = \sum_i G(v_i) \tag{6}$$

where $F(x)$ and $G(x)$ are some non-linear functions. In this limit we set $G(x) = |x|$. Since, the outputs of the feature neurons are derivatives of the Lagrangian (3), they are given by the sign

---

[2]There is also a border case possibility that the dynamics cycles without decreasing the energy (limit cycle), but this requires that the Hessian matrix in (4) has a zero mode everywhere along the trajectory. This border case possibility should be checked for a specific choice of the activation functions.

functions of their currents, which gives a set of binary variables that are denoted by $\sigma_i$

$$\sigma_i = g_i = g(v_i) = \frac{\partial L_v}{\partial v_i} = Sign[v_i] \tag{7}$$

Since $G(v_i) = |v_i|$ the only term that survives in the first square bracket in equation (2) is the one proportional to the input current $I_i$. The first term in the second bracket of equation (2) cancels the interaction term because of the steady state condition (5). Thus, in this limit the energy function (2) reduces to

$$E(t) = -\sum_{i=1}^{N_f} I_i \sigma_i - \sum_{\mu=1}^{N_h} F\Big(\sum_i \xi_{\mu i}\sigma_i\Big) \tag{8}$$

If there are no input currents $I_i = 0$ this is exactly the energy function for Dense Associative Memory from (Krotov & Hopfield, 2016). If $F(x) = x^n$ is a power function, the network can store $N_{\text{mem}} \sim N_f^{n-1}$ memories, if $F(x) = \exp(x)$ the network has exponential storage capacity Demircigil et al. (2017). If power $n = 2$ this model further reduces to the classical Hopfield network (Hopfield, 1982).

It is important to emphasize that the capacity estimates given above express the maximal number of memories that the associative memory can store given the dimensions of the input, but assuming no limits on the number of hidden neurons. In all the models considered in this work this capacity is also bounded by the number of those hidden neurons so that $N_{\text{mem}} \leq N_h$. With this constraint the capacity of model A with power function $F(x) = x^n$ should be written as

$$N_{\text{mem}} \sim \min(N_f^{n-1}, N_h) \tag{9}$$

In many practical applications (see examples in Section 4) the number of hidden neurons can be assumed to be larger than the bound defined by the dimensionality of the input space $N_f$. It is for this class of problems that Dense Associative Memories or modern Hopfield networks offer a powerful solution to the capacity limitation compared to the standard models of associative memory (Hopfield, 1982; 1984).

Lastly, for the class of additive models (6), which we call models A, the equation for the temporal evolution of the energy function reduces to

$$\frac{dE(t)}{dt} = -\tau_f \sum_{i=1}^{N_f} \Big(\frac{dv_i}{dt}\Big)^2 g(v_i)' - \tau_h \sum_{\mu=1}^{N_h} \Big(\frac{dh_\mu}{dt}\Big)^2 f(h_\mu)' \leq 0 \tag{10}$$

Thus, the condition that the Hessians are positive definite is equivalent to the condition that the activation functions $g(v_i)$ and $f(h_\mu)$ are monotonically increasing.

Additionally, in Appendix B, we show how standard continuous Hopfield networks (Hopfield, 1984) can be derived as a limiting case of the general theory (1,2).

## 3.2 Model B. Modern Hopfield Networks Limit and Attention Mechanism.

Models B are defined as models having contrastive normalization in the hidden layer. Specifically we are interested in

$$L_h = \log\Big(\sum_\mu e^{h_\mu}\Big), \quad \text{and} \quad L_v = \frac{1}{2}\sum_i v_i^2 \tag{11}$$

so that $L_v$ is still additive, but $L_h$ is not. Using the general definition of the activation functions (3) one obtains

$$f_\mu = \frac{\partial L_h}{\partial h_\mu} = \text{softmax}(h_\mu) = \frac{e^{h_\mu}}{\sum_\nu e^{h_\nu}}$$

$$g_i = \frac{\partial L_v}{\partial v_i} = v_i \tag{12}$$

Similarly to the previous case, consider the limit $\tau_h \to 0$, so that equation (5) is satisfied. In this limit the energy function (2) reduces to (currents $I_i$ are assumed to be zero)

$$E = \frac{1}{2}\sum_{i=1}^{N_f} v_i^2 - \log\Big(\sum_\mu \exp(\sum_i \xi_{\mu i} v_i)\Big) \tag{13}$$

This is exactly the energy function studied in (Ramsauer et al., 2020) up to additive constants (inverse temperature $\beta$ was assumed to be equal to one in this derivation). Notice that we used the notations from (Krotov & Hopfield, 2016), which are different from the notations of (Ramsauer et al., 2020). In the latter paper the state vector $v_i$ is denoted by $\xi_i$ and the memory matrix $\xi_{\mu i}$ is denoted by the matrix $\mathbf{X^T}$.

Making substitutions (12) in the first equation of (1), using steady state condition (5), and setting input current $I_i = 0$ results in the following effective equations for the feature neurons, when the memory neurons are integrated out

$$\tau_f \frac{dv_i}{dt} = \sum_{\mu=1}^{N_h} \xi_{i\mu} \text{softmax}\Big( \sum_{j=1}^{N_f} \xi_{\mu j} v_j \Big) - v_i \tag{14}$$

This is a continuous time counterpart of the update rule of (Ramsauer et al., 2020). Writing it in finite differences gives

$$v_i^{(t+1)} = v_i^{(t)} + \frac{dt}{\tau_f} \Big[ \sum_{\mu=1}^{N_h} \xi_{i\mu} \text{softmax}\Big( \sum_{j=1}^{N_f} \xi_{\mu j} v_j^{(t)} \Big) - v_i^{(t)} \Big] \tag{15}$$

which for $dt = \tau_f$ reduces to

$$v_i^{(t+1)} = \sum_{\mu=1}^{N_h} \xi_{i\mu} \text{softmax}\Big( \sum_{j=1}^{N_f} \xi_{\mu j} v_j^{(t)} \Big) \tag{16}$$

This is exactly the update rule derived in (Ramsauer et al., 2020), which, if applied once, is equivalent to the familiar dot-product attention (Bahdanau et al., 2014) and is also used in Transformer networks (Vaswani et al., 2017).

The derivation of this result in (Ramsauer et al., 2020) begins with the energy function for a Dense Associative Memory model with exponential interactions $F(x) = exp(x)$. Then it is proposed to take a logarithm of that energy (with a minus sign) and add a quadratic term in the state vector $v_i$ to ensure that it remains finite and the energy is bounded from below. While this is a possible logic, it requires a heuristic step - taking the logarithm, and makes the connection with Dense Associative Memories less transparent. In contrast, our derivation follows from the general principles specified by equations (1,2) for the specifically chosen Lagrangians.

It is also important to note, that the Hessian matrix for the hidden neurons has a zero mode (zero eigenvalue) for this limit of our model.

## 3.3 MODEL C. SPHERICAL MEMORY.

Models C are defined as having spherical normalization in the feature layer. We are not aware of a discussion of this class of associative memory models in the literature. Specifically,

$$L_h = \sum_\mu F(h_\mu), \quad \text{and} \quad L_v = \sqrt{\sum_i v_i^2} \tag{17}$$

so that $L_h$ is additive, but $L_v$ is not. Using the general definition of the activation functions (3) one obtains

$$f_\mu = F'(h_\mu)$$
$$g_i = \frac{\partial L_v}{\partial v_i} = \frac{v_i}{\sqrt{\sum_j v_j^2}} \tag{18}$$

Equations (1) for model C are given by ($I_i$ is assumed to be zero)

$$\begin{cases} \tau_f \frac{dv_i}{dt} = \sum_{\mu=1}^{N_h} \xi_{i\mu} f(h_\mu) - \alpha v_i \\ \tau_h \frac{dh_\mu}{dt} = \sum_{i=1}^{N_f} \xi_{\mu i} g_i - h_\mu \end{cases} \tag{19}$$

Notice, that since the Hessian matrix for the feature neurons has a zero mode proportional to $v_i$ in this model,

$$M_{ij} = \frac{\partial^2 L_v}{\partial v_i \partial v_j} = \frac{1}{\left(\sum_k v_k^2\right)^{3/2}} \left[\delta_{ij} \sum_l v_l^2 - v_i v_j\right], \quad \text{so that} \quad \sum_j M_{ij} v_j = 0, \qquad (20)$$

we can write an arbitrary coefficient $\alpha$, which can be equal to zero, in front of the decay term for the feature neurons. Taking the limit $\tau_h \to 0$ and excluding $h_\mu$ gives the effective energy

$$E(t) = -\sum_\mu F\left(\sum_i \xi_{\mu i} \frac{v_i}{\sqrt{\sum_j v_j^2}}\right) \qquad (21)$$

and the corresponding effective dynamical equations

$$\tau_f \frac{dv_i}{dt} = \sum_\mu \xi_{i\mu} f\left[\sum_j \xi_{\mu j} \frac{v_j}{\sqrt{\sum_k v_k^2}}\right] - \alpha v_i \qquad (22)$$

It is also important to notice that the activation function $g_i$ that appears in equation (18) implements a canonical computation of divisive normalization widely studied in neuroscience (Carandini & Heeger, 2012). Divisive normalization has also been shown to be beneficial in deep CNNs and RNNs trained on image classification and language modelling tasks (Ren et al., 2016).

## 4 A FEW EXAMPLES OF LARGE ASSOCIATIVE MEMORY PROBLEMS

In this section we provide some examples of problems in AI and biology which may benefit from thinking about them through the lens of associative memory.

**Pattern memorization.** Consider a small gray scale image $64 \times 64$ pixels. If one treats the intensity of each pixel as an input to a feature neuron the standard associative memory (Hopfield, 1982) would be able to only memorize approximately $0.14 \cdot 4096 \approx 573$ distinct patters. Yet, the number of all possible patterns of this size that one can imagine is far bigger. For instance, Kuzushiji-Kanji dataset (Clanuwat et al., 2018) includes over 140,000 characters representing 3832 classes with most of the characters recognizable by humans. A well educated Japanese person can recognize about 3000-5000 character classes, which means that those classes are represented in his/her memory. In addition, for many characters a person would be able to complete it if only a portion of that character is shown. Moreover, possible patterns of $64 \times 64$ pixels are not necessarily Kanji characters, but also include digits, smileys, emojis, etc. Thus, the overall number of patterns that one might want to memorize is even bigger.

In the problem of **immune repertoire classification**, considered in (Widrich et al., 2020), the number of immune repertoire sequences (number of memories in the modern Hopfield network) is $N \gg 10000$, while the size of the sequence embedding dimension $d_k = 32$, or $N_f = 32$ using the notations of this current paper. The ability to solve this problem requires the associative memory used in the aforementioned paper to have a storage capacity much larger than the dimensionality of the feature space.

**Cortical-hippocampal system.** The hippocampus has long been hypothesised to be responsible for formation and retrieval of associative memories, see for example (Rolls, 2018; Treves & Rolls, 1994). Damage to the hippocampus results in deficiencies in learning about places and memory recognition visual tasks. For instance (Parkinson et al., 1988) reports deficiencies in object-memory tasks, which require a memory of an object and the place where that object was seen. One candidate for associative memory network in the hippocampus is the CA3 area, which consists of a large population of pyramidal neurons, approximately $3 \cdot 10^5$ in the rat brain (Amaral & Witter, 1989), and $2.3 \cdot 10^6$ (Seress, 1988) in human, in conjunction with an inhibitory network that keeps the firing rates under control. There is a substantial recurrent connectivity among the pyramidal neurons (Rolls, 2018), which is necessary for an associative memory network. There are also several classes of responses of those neurons in behaving animals, one class being place cells (O'Keefe & Dostrovsky, 1971). In addition to place cells (Ferguson et al., 2011) report existence of cells in the hippocampus that do not respond in experiments designed to drive place cells, but presumably are useful for other tasks. One possible way of connecting the mathematical model proposed in this paper with

the existing anatomical network in the brain is to assume that some of the pyramidal cells in CA3 correspond to the feature neurons in our model, while the remaining pyramidal cells are the memory neurons. For example, place cells are believed to emerge as a result of aggregating inputs from the grid cells and environmental features, e.g. landmark objects, environment boundaries, visual and olfactory cues, etc., (Moser et al., 2015). Thus, it is tempting to think about them as memory neurons (which aggregate information from feature neurons to form a stable memory) in the proposed model.

Another area of the hippocampus potentially related to the mathematical model described in this paper is the area CA1, which, in addition to receiving inputs from CA3, also receives inputs directly from the entorhinal cortex, and projects back to it. In this interpretation pyramidal cells of the CA1 would be interpreted as the memory neurons in our mathematical model, while the cells in the layer III of the entorhinal cortex would be the feature neurons. The feedback projections from CA1 go primarily to layer V of the entorhinal cortex (Rolls, 2018), but there are also projections to layers II and III (Witter et al., 2017). While it is possible to connect the proposed mathematical model of Dense Associative Memory with existing networks in the hippocampus, it is important to emphasize that the hippocampus is involved in many tasks, for example imagining the future (Hassabis et al., 2007), and not only in retrieving the memories about the past. For this reason it is difficult at present to separate the network motifs responsible for memory retrievals from the circuitry required for other functions.

**Human color vision** has three dimensions so that every color sensation can be achieved by mixing three primary lights (Mollon, 2003). From the neuron's perspective every color is detected by three kinds of cone photoreceptors ($N_f = 3$) in the retina, so that the degree of excitation of each photoreceptor is described by a continuous number. Most people know many colors with names for them (e.g. red, orange, yellow, green, blue, indigo, violet, pink, lavender, copper, gold, etc.), descriptions for others. e.g. "the color of the sky". Experimentally, humans can distinguish about $10^6$ different colors (Masaoka et al., 2013), although may not be able to "memorize" all of them. See also (Meister, 2015) for the discussion of this problem. Thus, if one thinks about this system as an associative memory for color discrimination, the model of (Hopfield, 1982) and its extensions with $O(N_f)$ storage capacity would be inadequate since they can only "remember" a few colors. It is important to emphasize that the memories of the colors are stored in higher areas of the brain, while the color sensation is conveyed to the brain through the cone cells in the retina. Thus, in this example there are many intermediate neurons and synapses between in feature neurons and memory neurons. For this reason it is only appropriate to think about this example as a direct associative memory if all these intermediate neurons and synapses are integrated out from this system.

## 5    DISCUSSION AND CONCLUSIONS

We have proposed a general dynamical system and an energy function that has a large memory storage capacity, and, at the same time, is manifestly describable in terms of two-body synaptic connections. From the perspective of neuroscience it suggests that Dense Associative Memory models are not just mathematical tools useful in AI, but have a degree of biological plausibility similar to that of the conventional continuous Hopfield networks (Hopfield, 1984). Compared to the latter, these models have a greater degree of psychological plausibility, since they can store a much larger number of memories, which is necessary to explain memory-based animal behavior.

We want to emphasize that the increase in the memory storage capacity that is achieved by modern Hopfield networks is a result of unfolding the effective theory and addition of (hidden) neurons. By adding these extra neurons we have also added synapses. Coming back to the information counting argument that we presented in the introduction, the reason why these unfolded models have a larger storage capacity than the conventional Hopfield networks with the same number of input neurons is because they have more synapses, but each of those synapses has the same information capacity as in the conventional case.

From the perspective of AI research our paper provides a conceptually grounded derivation of various associative memory models discussed in the literature, and relationships between them. We hope that the more general formulation, presented in this work, will assist in the development of new models of this class that could be used as building components of new recurrent neural network architectures.

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

ACKNOWLEDGEMENTS

We are thankful to J. Brandstetter, S. Hochreiter, M. Kopp, D. Kreil, H.Ramsauer, D. Springer, and F. Tang for useful discussions.

## APPENDIX A

In this appendix we show a step by step derivation of the change of the energy function (2) under dynamics (1). Time derivative of the energy function can be expressed through time derivatives of the neuron's activities $v_i$ and $h_\mu$ (the input current $I_i$ is assumed to be time-independent in the calculation below). Using the definition of the functions $f_\mu$ and $g_i$ in (3) one can obtain

$$
\begin{aligned}
\frac{dE}{dt} &= \sum_{i,j} (v_i - I_i) \frac{\partial^2 L_v}{\partial v_i \partial v_j} \frac{dv_j}{dt} + \sum_{\mu,\nu} h_\mu \frac{\partial^2 L_h}{\partial h_\mu \partial h_\nu} \frac{dh_\nu}{dt} \\
&\quad - \sum_{\mu,\nu} \frac{dh_\nu}{dt} \frac{\partial^2 L_h}{\partial h_\nu \partial h_\mu} \Big( \sum_i \xi_{\mu i} g_i \Big) - \sum_{i,j} \frac{dv_j}{dt} \frac{\partial^2 L_v}{\partial v_j \partial v_i} \Big( \sum_\mu \xi_{i\mu} f_\mu \Big) = \\
&\quad - \sum_{i,j} \frac{dv_j}{dt} \frac{\partial^2 L_v}{\partial v_j \partial v_i} \Big[ \sum_\mu \xi_{i\mu} f_\mu + I_i - v_i \Big] - \sum_{\mu,\nu} \frac{dh_\nu}{dt} \frac{\partial^2 L_h}{\partial h_\nu \partial h_\mu} \Big[ \sum_i \xi_{\mu i} g_i - h_\mu \Big] = \\
&\quad - \tau_f \sum_{i,j=1}^{N_f} \frac{dv_i}{dt} \frac{\partial^2 L_v}{\partial v_i \partial v_j} \frac{dv_j}{dt} - \tau_h \sum_{\mu,\nu=1}^{N_h} \frac{dh_\mu}{dt} \frac{\partial^2 L_h}{\partial h_\mu \partial h_\nu} \frac{dh_\nu}{dt} \leq 0
\end{aligned}
\tag{23}
$$

In the last equality sign the right hand sides of dynamical equations (1) are used to replace expressions in the square brackets by the corresponding time derivatives of the neuron's activities. This completes the proof that the energy function decreases on the dynamical trajectory described by equations (1) for arbitrary time constants $\tau_f$ and $\tau_h$ provided that the Hessians for feature and memory neurons are positive semi-definite.

## APPENDIX B. THE LIMIT OF STANDARD CONTINUOUS HOPFIELD NETWORKS.

In this section we explain how the classical formulation of continuous Hopfield networks (Hopfield, 1984) emerges from the general theory (1,2). Continuous Hopfield networks for neurons with graded response are typically described by the dynamical equations

$$
\tau_f \frac{dv_i}{dt} = \sum_{j=1}^{N_f} T_{ij} g_j - v_i + I_i
\tag{24}
$$

and the energy function

$$
E = -\frac{1}{2} \sum_{i,j=1}^{N_f} T_{ij} g_i g_j - \sum_{i=1}^{N_f} g_i I_i + \sum_{i=1}^{N_f} \int^{g_i} g^{-1}(z) dz
\tag{25}
$$

where, as in Section 3.1, $g_i = g(v_i)$, and $g^{-1}(z)$ is the inverse of the activation function $g(x)$.

According to our classification, this model is a special limit of the class of models that we call models A, with the following choice of the Lagrangian functions

$$
L_v = \sum_{i=1}^{N_f} \int^{v_i} g(x) dx, \quad \text{and} \quad L_h = \frac{1}{2} \sum_{\mu=1}^{N_h} h_\mu^2
\tag{26}
$$

that, according to the definition (3), lead to the activation functions

$$
g_i = g(v_i), \quad \text{and} \quad f_\mu = h_\mu
\tag{27}
$$

Similarly to Section 3.1, we integrate out the hidden neurons to demonstrate that the system of equations (1) reduces to the equations on the feature neurons (24) with $T_{ij} = \sum_{\mu=1}^{N_h} \xi_{\mu i} \xi_{\mu j}$. The general expression for the energy (2) reduces to the effective energy

$$E = -\frac{1}{2} \sum_{i,j=1}^{N_f} T_{ij} g_i g_j - \sum_{i=1}^{N_f} g_i I_i + \sum_{i=1}^{N_f} \left( v_i g_i - \int^{v_i} g(x) dx \right) \tag{28}$$

While the first two terms in equation (25) are the same as those in equation (28), the third terms look superficially different. In equation (28) it is a Legendre transform of the Lagrangian for the feature neurons, while in (25) the third term is an integral of the inverse activation function. Nevertheless, these two expressions are in fact equivalent, since the derivatives of a function and its Legendre transform are inverse functions of each other. The easiest way to see that these two terms are equal explicitly is to differentiate each one with respect to $v_i$. The results of these differentiations for both expressions are equal to $v_i g(v_i)'$. Thus, the two expressions are equal up to an additive constant. This completes the proof that the classical Hopfield network with continuous states (Hopfield, 1984) is a special limiting case of the general theory (1, 2).

