# OpenReview forum: "Large Associative Memory Problem in Neurobiology and Machine Learning"
_ICLR.cc/2021/Conference — ICLR 2021 Poster_

### Official Review · AnonReviewer1 · 2020-10-25
**Would benefit from better explanation of tradeoff and clearer motivation of energy function**

**Rating:** 7
**Confidence:** 3

**Review:**

The authors proposed a dynamical system that unifies several associative memory models, including the classical Hopfield network and two recently proposed modern Hopfield networks. The dynamical system is described as interactions between two groups of neurons (feature and memory neurons), providing a more biological interpretation of modern Hopfield networks. The proposed system reduces to different associative memory models by choosing different generalized activation functions, each of which maps inputs to a group of neurons into output activity. This manuscript provides sufficient details for understanding, its derivations are correct, and its results are useful in bringing modern Hopfield networks closer to biology.

### Major:

(1)	My main concern is that the tradeoff of the modern Hopfield networks discussed is not clearly stated.

For example, the manuscript at multiple places described modern Hopfield networks as having large or huge memory capacity with respect to $N_f$. I understand that these statements have been used in the literature. However, since the authors are explicitly trying to interpret modern Hopfield networks as network of neurons, it would be appropriate to discuss the memory capacity with respect to both $N_f$ and $N_h$. For example in model B, if I understood correctly, the memory capacity scales linearly with $N_h$.

Related to the previous point, the number of memory neurons needed should be clearly described. Is it the number of memory patterns stored across all models?

(2)	The energy function should be better motivated.

In equation 2, the authors introduce an energy function containing two Lagrangian functions, and demonstrate that it reduces to energy functions from several previous models when using different choices of Lagrangian functions. While this is intriguing, it can make the reader feels like this energy function came out of nowhere and magically unifies multiple models. I believe the reader would have a better understanding if the authors provide a stronger motivation for the energy function. For example, is the energy function constructed by explicitly trying to connect the Krotov & Hopfield 2016 model with the Ramsauer 2020 model?

### Additional major points:

(3)	I think the authors should offer richer references to the literature. Here are some examples:

In the intro, “a cortical synapse stores no more than one or two bits of information” should be followed by a reference since this is not common knowledge. For example, Bartol 2015 eLife estimates a lower bound of 4.7 bits per synapse.

In p.4, the authors mention BERT-like system without citing the BERT paper.

The activation function in Eq. 17 is a form of divisive normalization widely studied in neuroscience (Carandini & Heeger 2012).

(4)	I find it somewhat misleading to refer to the update rule in Eq. 15 as the “attention mechanism in Transformer networks”. I understand this is what Ramsauer et al 2020 said as well, but I think the update rule in Eq. 15 should be simply referred to as dot-product attention, which is used in machine learning at least as early as Bahdanau, Cho, Bengio 2014 and in many settings besides Transformers.

### Minor:

P.2 “integrates our some of the degrees of freedom”  “integrates out…”

In P.1, the sentence “a small part of a high-resolution photograph may contain only 1000 pixels, but the number of describable “objects” which might occur in such a fragment is far larger” is a bit confusing.


-----------------------------------
Post revision update:
All concerns addressed. Score updated.

---

> ### Author Response · Authors · 2020-11-18
> **Response to AnonReviewer1**
>
> Thank you for the nice comment: “This manuscript provides sufficient details for understanding, its derivations are correct, and its results are useful in bringing modern Hopfield networks closer to biology.” Below we address the questions raised:
>
> 1. We have added a discussion to the revised manuscript of the capacity as a function of $N_h$ (paragraph around expression 9). Indeed the capacity scales linearly with $N_h$. This is not an issue, however, since our model tackles problems where one needs to store many more memories than the number of feature neurons (number of hidden neurons $N_h$ is assumed to be large). We have added the discussion to Introduction of specific examples from biological and machine learning systems where the number of feature neurons is fixed, yet the number of needed “memories” is much bigger than $N_f$. In our model the number of stored  memory patterns is bounded by the number of memory neurons (equation 9).  We hope that these modifications address your concern.
>
> 2.  (The energy function should be better motivated): Our logic was to allow a sufficient flexibility in the dynamical equations (1) so that they can be reduced to both (Krotov & Hopfield, 2016) and (Ramsauer et al., 2020) in limiting cases, subject to the “biological constraint” that the synapses should only be pair-wise. As is always the case, finding a sufficiently simple Lyapunov function (which is not unique - any increasing function of equation 2 for example would also be a Lyapunov function) requires some level of trials and errors. We have added a few sentences before equation (2) to better motivate it. We hope that this modification helps the reader to better follow our reasoning.
>
> 3. (I think the authors should offer richer references to the literature): Thank you very much for the proposed references! We have added all of them to the revised manuscript. We completely agree that it is better to call the right hand side of the update rule for model B “dot-product attention”. Discussion of divisive normalization is added at the end of section 3.3. In general, we have significantly expanded the cited literature in the revised version of the paper.
>
> 4. (Minor): The typo is corrected and the confusing sentence rephrased. Thanks for pointing these out!

---

> > ### Comment · AnonReviewer1 · 2020-11-18
> > **Concerns addressed, score updated**
> >
> > Thanks to the author's revision, my concerns are all addressed. I have updated the score accordingly.

---

> > > ### Author Response · Authors · 2020-11-24
> > > **Thanks**
> > >
> > > Thank you very much!

---

### Official Review · AnonReviewer4 · 2020-10-27
**Good paper, though biological plausibility angle overstated**

**Rating:** 8
**Confidence:** 3

**Review:**

This paper presents a novel class of associative memory models. The model is expressed as a network with two-body interactions (synapses) and a well-defined energy function, and it is shown to generalise and unify several existing approaches (Hopfield Networks, Dense Associative Memories and Modern Hopfield Networks). Besides its theoretical and computational properties, the model is presented as being more biologically valid/plausible than some of the existing approaches it generalizes.

Overall the paper is solid, well written and highly relevant to the ICLR community. My initial recommendation is to accept. I only have a few nitpicks concerning the "biologically plausible" angle.

More specifically, while I agree that the proposed model is *more* plausible than some of the other approaches discussed, its absolute level of biological plausibility remains limited. The authors recognise this point and devote a paragraph to discussing it at the end of section 2 ("For the purposes of this paper we defined biological plausibility as…"), but it seems odd to have this passage buried at the end of the mathematical derivation, rather than up front in the introduction. I suggest that this passage is moved forward to a more prominent location.

Furthermore, given that this is essentially a paper on the theory of abstract associative memory systems, the emphasis given to the biological angle in the title seems eccessive, and the choice of the word "neurobiology" somewhat puzzling. In my opinion the title would be a better description of the content of the paper if the reference to biology was toned down.

Finally, in the introduction: "typical synapses are not highly reliable, and a cortical synapse stores no more than one or two bits of information". While I agree with the general spirit of the observation that synapses are not typically very reliable, some source should provided to back up the quantitative statement about synaptic storage capacity. I am not a specialist on the matter, but this seems surprising in light of work showing that the capacity of hippocampal synapses can be up to 3 to 5 bits (Bartol et al 2015, Bromer et al 2018).

### References

Bartol Jr, T. M., Bromer, C., Kinney, J., Chirillo, M. A., Bourne, J. N., Harris, K. M., & Sejnowski, T. J. (2015). Nanoconnectomic upper bound on the variability of synaptic plasticity. Elife, 4, e10778.

Bromer, C., Bartol, T. M., Bowden, J. B., Hubbard, D. D., Hanka, D. C., Gonzalez, P. V., … & Sejnowski, T. J. (2018). Long-term potentiation expands information content of hippocampal dentate gyrus synapses. Proceedings of the National Academy of Sciences, 115(10), E2410-E2418.

----

Post revision update: my concerns have all been addressed.

---

> ### Author Response · Authors · 2020-11-18
> **Response to AnonReviewer4**
>
> Thank you very much for the enthusiastic evaluation and valuable feedback! We are glad to hear that "the paper is solid, well written and highly relevant to the ICLR community". Below we address the comments:
>
> 1. (More specifically, while I agree that the proposed model is more plausible than…): Indeed this paragraph better belongs to the introduction. We have moved it there in the revised version. Thanks for the suggestion!
>
> 2. (Furthermore, given that this is essentially a paper on the theory of abstract associative memory systems, the emphasis given to the biological angle in the title seems eccessive…): Thanks for this feedback. We have toned down the “biological angle” throughout the revised manuscript, but decided to keep the word “neurobiology” in the title. The requirement that the model is only allowed to have pair-wise synapses is motivated by the biological constraints. We have also expanded the introduction with a few examples of specific systems in biology and machine learning where it is beneficial to have a large number of memories compared to the number of feature neurons. We hope that the combination of these changes addresses your concern.
>
> 3. (Finally, in the introduction: "typical synapses are not highly reliable…): Thanks for the references! We have expanded the discussion of this issue and added the references that you proposed, as well as other related literature and clarifications of this claim. Please see the footnote on page 1.

---

> > ### Comment · AnonReviewer4 · 2020-11-19
> > **Thanks for the response, I find the new introduction clearer**
> >
> > Thank you for the response, I find that the new introduction does a better job at motivating the biological relevance of the work. I have updated my official review.

---

> > > ### Author Response · Authors · 2020-11-24
> > > **Thanks**
> > >
> > > Thank you for your time to evaluate our revision!

---

### Official Review · AnonReviewer2 · 2020-10-28
**Theoretically sound but the significance is unclear to this reader**

**Rating:** 6
**Confidence:** 4

**Review:**

This paper provides a mechanism by which recent extensions to the classic Hopfield network model, which as written involve many-body interactions, can be implemented by a more biologically plausible network that uses only two-body synaptic interactions (but more neurons).

Pros:

-- The derivations are sound and well-explained, from what I can tell
-- Given that Hopfield networks are an important model in the neuroscience community, it is nice to see these extensions to them brought back in to the realm of biological implementation

Cons:

-- My main concern with this paper is that I feel like it has not established its significance to neuroscience modelers -- and since the paper is primarily concerned with connecting existing algorithms to biologically plausible implementations, rather than introducing new algorithms, I think it is important to do so.  Dense associative memories and modern Hopfield networks are appealing because of their ability to store memories beyond the O(N) scaling limit of regular Hopfield networks (N = # of neurons).  But in these two-body implementations, the models once again have O(N) scaling in the total number of neurons due to the addition of hidden neurons.  Given this, what is the advantage of these models over the standard Hopfield network?  Are they still more efficient in some way?  More robust?  Do they provide faster recall?  I think a thorough comparison along these lines would be valuable and without it, the paper is less compelling to me.

-- The connection to attention mechanisms feels a bit off in that the equivalence only holds when the update rule is applied exactly once and no more times (I realize this was first shown in prior work).  But given that this paper is attempting to unify several mechanisms under a common framework, it feels like the paper should concern itself with models that really are variants of that framework, rather than introducing additional modifications like cutting off the dynamics early.

---

> ### Author Response · Authors · 2020-11-18
> **Response to AnonReviewer2**
>
> Thank you for your feedback. We are glad to hear that the derivations are written clearly. Below we address the questions raised:
>
> 1. (My main concern with this paper is that I feel like it has not established its significance to neuroscience modelers…): We have significantly expanded Introduction section and added three examples of problems in which the number of memories required for the proper functionality should be significantly larger than the number of neurons in the feature space. These are examples of the kinds of problems our approach tackles. The main advantage of Modern Hopfield Networks compared to classical Hopfield Networks is the ability to store a significantly larger number of memories than the number of feature neurons, not the overall number of neurons. It is true that the number of  memories is bounded by the number of memory neurons (we have added an explicit paragraph to the revised manuscript emphasizing this point - around equation 9), but this is not a drawback of the approach. As we explain in the “information counting argument” in order to expand the memory storage capacity, one would have to introduce a sufficiently large number of extra neurons. But this is OK, since there are plenty of hidden neurons in the brain. It is the feature neurons that are limited in numbers. Also please notice that the standard network from (Hopfield, 1982) and its $O(N_f)$ extensions would not be able to successfully solve any of the three problems that we discuss in the revised version of the Introduction. For instance, it would not be able to solve well the multiple instance learning problem pertaining to the immune repertoire classification, as explained in (Widrich et al., 2020). At the same time, modern Hopfield networks can solve these problems. This is their main advantage, compared to the standard Hopfield nets.
>
> 2. (The connection to attention mechanisms feels a bit off in that the equivalence only holds when the update rule is applied exactly once and no more times): This is correct - the update rule Eq (16) is more general than the attention mechanism. It only reduces to the latter if applied only once. It is an interesting research question, already raised in (Ramsauer et al., 2020) and not discussed in our present submission, whether extending the standard attention mechanism to multiple step updates, like in equations 14-16, would improve the Transformers’ performance.

---

> > ### Comment · AnonReviewer2 · 2020-11-18
> > **Response**
> >
> > Thanks for your response.  I am confused by your point #1.  Why should scaling in the number of "feature" neurons be important?  Astandard Hopfield network can also take advantage of a number of neurons greater than the initial feature dimensionality, for example by performing a random expansion from the initial feature space prior to the Hopfield network.   Is there reason to think that such an approach would be outperformed by modern Hopfield networks?  There very well could be, but if so I am missing it.
> >
> > Relatedly, I find the statement "there are plenty of hidden neurons in the brain. It is the feature neurons that are limited in numbers" a bit confusing, as any neurons being used to represent a stimulus / concept (e.g. those involved in a dimensionality expansion as above) can just as well be thought of as "feature neurons."

---

> > > ### Author Response · Authors · 2020-11-18
> > > **Response to AnonReviewer2**
> > >
> > > Thank you for your comment. Are you proposing the following construction? Take an $N_f$-dimensional feature vector $v_i^{t=0}$ and expand it to a $N_{\text{exp}}$-dimensional vector $q_\alpha^{t=0}$ using a random matrix $X_{\alpha i}$. Index $t$ refers to the discrete time steps of the RNN. This way $q_\alpha^{t=0} = p\Big(\sum\limits_i X_{\alpha i} v_i^{t=0}\Big)$, where $p(\cdot)$ is a non-linear function. Then apply standard Hopfield network dynamics $q_\alpha^{t+1} = \text{SHN}(q_\alpha^{t})$.
> > >
> > > If this is what you mean, then such a system would not be a proper recurrent network for the feature neurons $v_i$, because after the expanded state vector $q_\alpha$ is updated by the standard Hopfield network it would be impossible to find the corresponding state vector $v_i$. For example after one step update one would need to solve the following system of $N_{\text{exp}}$ equations on $N_f$ variables $q_\alpha^{t=1} = p\Big(\sum\limits_i X_{\alpha i} v_i^{t=1}\Big)$. Given that $N_{\text{exp}}\gg N_f$ such a system in general would not be solvable. For this reason, this construction would not be an associative memory for the feature neurons $v_i$.

---

### Official Review · AnonReviewer3 · 2020-10-28
**Associative Memory in Bio and ML**

**Rating:** 7
**Confidence:** 3

**Review:**

Summary: In the current work, the authors describe a novel memory structure, and mathematically show how this is a superset of previously published models. The paper could be of interest to anyone investigating the theoretical side of learning and activity rules.

Strong Points: Unifying the multiple discussed architectures, as well as relating back to RBM and classical Hopfield networks is a nontrivial task. Additionally, the use of integrator units (eq 1) provides a clear path how such an energy optimizing network might be implemented in biological networks (eg: LIF neurons).

Weak Points: Biological plausibility is discussed at several points in the paper, specifically regarding the number of units involved in an interaction. However, this plausibility doesn't influence the functional design of the network, and thus feels extraneous.

Additional Comments:
Regarding the above: If the feature units are to represent distributed patterns (eg: BERT features), then it should be possible for more than 2 abstract units to interact at a single "synapse", so long as the interaction of the actual units (neurons or RELU units) has only two members?
Are the memory capacity estimates (directly following eq 8) only applicable when the input current is 0? If so, is it possible to derive more general estimates of capacity?

---

> ### Author Response · Authors · 2020-11-18
> **Response to AnonReviewer3**
>
> Thank you for the positive feedback, we are glad that the message regarding the relationship of our model to RBMs and classical Hopfield networks got across.
>
> Although we agree that the core idea of our model is in its mathematical structure, we believe that the biological motivation is important here since it is this motivation that makes us look for a mathematical description of large associative memory models in terms of only pair-wise synapses. Following your feedback and the suggestion of [Rev 4] we have decided to move the discussion of the biological plausibility of two-body synapses from section 2 to the introduction. We have also added to the Introduction the discussion of biological and AI systems which may benefit from the interpretation based on associative memory.
>
> 1. (If the feature units are to represent distributed patterns (eg: BERT features), then it should be possible for more than 2 abstract units to interact at a single "synapse", so long as the interaction of the actual units (neurons or RELU units) has only two members?): In our model arbitrary number of feature neurons can interact with each other in the effective description (when the memory neurons are excluded), but in the microscopic theory these interactions are mediated through pair-wise interactions with memory neurons.
>
> 2. (Are the memory capacity estimates (directly following eq 8) only applicable when the input current is 0? If so, is it possible to derive more general estimates of capacity?): In our model the storage of the memories is accomplished by the interactions between the feature neurons and the memory neurons. The input current, which only couples to the feature neurons, provides a guidance to the system which basin of attraction it would eventually converge to, but does not influence the capacity of the model, provided that this current is not too large. For example, consider model A described by the energy function Eq (8). In this formula the function $F(x) = x^n$ or $exp(x)$ is a very rapidly growing function of the overlap between the memories and the state of the network. Imagine for simplicity that the currents, the memories $\xi_{\mu i}$ and the state variable $\sigma_i$ are all binary variables $\{\pm 1\}$. If the state variable $\sigma_i$ is perfectly aligned with one of the memories in this formula the leading contribution, coming from the second term in Eq (8), will be $N_f^n$, or $exp(N_f)$ for the two choices of the activation function. At the same time, the first term can only be of the order of $N_f$ (when $I_i$ is perfectly aligned with $\sigma_i$). Thus, in the interesting regime, when this system works as an associative memory, the second term dominates the first one. Thus, in this limit, the addition of the input current does not change the capacity.
> In the opposite limit, when the current $I_i$ is huge and outweighs the second term, the model will only have one local minimum with $\sigma_i = I_i$. There is also a complicated intermediate case when the two terms have approximately equal importance. We hope to investigate this intermediate regime in the future. The main focus of our current work, however, is the case when the second term in Eq (8) is dominant.

---

### Decision · Program_Chairs · 2021-01-07
**Final Decision**

**Decision:**

Accept (Poster)

**Comment:**

This paper attends to the problem of how to implement dense associative memories (i.e. modern Hopfield networks) using only two-body synapses. This is interesting because modern Hopfield networks have much higher capacity, but at face value, they require synapses with cubic interactions between neurons, which to the best of our knowledge, is not a common feature in neurophysiology (though it should be noted: it is not by any means impossible from a physiological perspective to have cubic interactions at synapses, see e.g. Halassa, M. M., Fellin, T., & Haydon, P. G. (2007). The tripartite synapse: roles for gliotransmission in health and disease. Trends in molecular medicine, 13(2), 54-63.).

The authors show how the use of a layer of hidden neurons, akin to a restricted Boltzmann machine architecture, coupled with the right energy function, can be used to recover dense associative memory models using only two-body synapses. They also demonstrate how this connects to recent work on the relationship between attention mechanisms in modern ML models and Hopfield network dynamics.

Overall, the reviewers were positive on this paper. The most common critique related to the question of "biological plausibility". The authors addressed these concerns by adding some more recognition as to the lack of biological plausibility and more discussions of the relevance to neuroscience. To be candid with the authors, if the goal is indeed to make a more biologically plausible model of modern Hopfield networks, than a fair bit more work would be needed to connect the paper to biology well. As it stands, the only connection is the shift to two-body synapses by using hidden neurons, but this provides limited insight for most neuroscientists, as noted by Reviewer 2. Also, some of the biological examples provided seem strained (e.g. the colour memory example, where there is no physiological reason to posit that we store colour memories using our retina, or the MNIST example, since there is no reason to suppose that animals can memorise thousands of specific MNIST images). But overall, the critique regarding biological plausibility was attended to. The other concerns raises were also largely addressed.

Given the interesting contributions from this paper, the overall positive reviews, and the decent job at addressing reviewer concerns, the AC believes that this paper should certainly be accepted. A decision of "Accept (Poster)" seems appropriate, though (as opposed to an oral or spotlight), given the lack of biological connections in a paper with a stated goal of achieving a more biologically realistic model.